# Learning induces coordinated neuronal plasticity of metabolic demands and functional brain networks

Sebastian Klug [1,5], Godber M. Godbersen [1,5], Lucas Rischka [1], Wolfgang Wadsak [2,3],
Verena Pichler [2,4], Manfred Klöbl[1], Marcus Hacker [2], Rupert Lanzenberger [1] & Andreas Hahn [1✉]

The neurobiological basis of learning is reflected in adaptations of brain structure, network organization and energy metabolism. However, it is still unknown how different neuroplastic mechanisms act together and if cognitive advancements relate to general or task-specific changes. Therefore, we tested how hierarchical network interactions contribute to improvements in the performance of a visuo-spatial processing task by employing simultaneous PET/MR neuroimaging before and after a 4-week learning period. We combined functional PET and metabolic connectivity mapping (MCM) to infer directional interactions across brain regions. Learning altered the top-down regulation of the salience network onto the occipital cortex, with increases in MCM at resting-state and decreases during task execution. Accordingly, a higher divergence between resting-state and task-specific effects was associated with better cognitive performance, indicating that these adaptations are complementary and both required for successful visuo-spatial skill learning. Simulations further showed that changes at resting-state were dependent on glucose metabolism, whereas those during task performance were driven by functional connectivity between salience and visual networks. Referring to previous work, we suggest that learning establishes a metabolically expensive skill engram at rest, whose retrieval serves for efficient task execution by minimizing prediction errors between neuronal representations of brain regions on different hierarchical levels.

[1] Department of Psychiatry and Psychotherapy, Medical University of Vienna, Vienna, Austria. [2] Department of Biomedical Imaging and Image-guided Therapy, Division of Nuclear Medicine, Medical University of Vienna, Vienna, Austria. [3] Center for Biomarker Research in Medicine (CBmed), Graz, Austria. [4] Department of Pharmaceutical Sciences, Division of Pharmaceutical Chemistry, University of Vienna, Vienna, Austria. [5]These authors contributed equally: Sebastian Klug, Godber M Godbersen. ✉email: andreas.hahn@meduniwien.ac.at

Learning is a fundamental cognitive process that allows an organism to adapt to its environment throughout its entire lifetime. As neuroscientific research has moved beyond the behavioral level to examine the underlying neurobiological mechanisms, numerous structural, functional, and molecular aspects of learning have been demonstrated.

Human magnetic resonance imaging (MRI) studies have revealed learning-induced neuroplastic changes in gray and white matter structure[1] as well as in functional networks that undergo dynamic reconfigurations[2]. These consistently showed that interactions of higher-order brain networks of cognitive control, such as cingulo-opercular, salience, and fronto-parietal networks, among each other[3] and with lower-level visual areas[4] predict learning-related gain in task efficiency[5]. On the other hand, positron emission tomography (PET) imaging with the radiolabeled glucose analogue [18F]FDG further showed that brain areas involved in a visuo-spatial task performance also undergo metabolic adaptations after learning, indicating a more cost-effective use of metabolic resources[6].

However, most of the previous work only employed a single imaging modality at the same time, thus impeding to draw conclusions about how the different parameters of brain function act together in the process of learning. In addition, neuroplastic effects were investigated either in a general manner at resting-state (e.g., gray and white matter structure[1], network adaptations) or specifically during task execution (e.g., metabolic demands[6], neuronal activation), while the direct comparison between the two states largely remains missing[7]. In sum, it is not clear whether intrinsic resting-state or task-related effects drive the improvement in cognitive performance after learning. Furthermore, the interaction between different indices of brain function and network adaptations is poorly understood.

The application of functional PET (fPET)[8] in the framework of metabolic connectivity mapping (MCM) provides a valuable approach to address both of these open questions. MCM combines MRI-derived functional connectivity and glucose metabolism obtained with [18F]FDG PET, thereby enabling the computation of directional connectivity[9]. The underlying rationale is that the integration of metabolic information identifies the target region of a connection since the majority of energy demands emerge post-synaptically[10–12]. The two imaging parameters are also tightly linked on a physiological basis through glutamate-mediated processes that occur upon neuronal activation. Glutamate release increases cerebral blood flow via neurovascular coupling[13,14], which in turn affects the blood oxygen level-dependent (BOLD) signal used for the assessment of functional connectivity. On the other hand, glutamate release also triggers glucose uptake into neurons[15] and astrocytes[16], to meet increased energy demands for the reversal of ion gradients[11,17,18]. MCM thus constitutes a validated framework to investigate the associations of glucose metabolism and functional connectivity and decipher hierarchical interactions across brain regions by assigning directionality to connections. For an in-depth discussion on the rationale and the underlying biological mechanisms of MCM the reader is referred to the previous work[9,19]. Using MCM, we have recently demonstrated that first-time performance of a cognitive task strengthened the interplay of functional connectivity and glucose metabolism, specifically for feedforward connections to higher-order cognitive processing areas[19]. However, the corresponding effects induced by prolonged training of a task remain unknown.

In the current work we aimed to address the open questions outlined above, namely (i) the interaction of training-induced changes between functional connectivity and glucose metabolism, (ii) the neurobiological contributions of resting-state and task-specific effects that drive improvements in cognitive performance, and (iii) the hierarchical interplay across brain regions involved in the learning process. We investigated learning-induced neuronal adaptations in functional brain networks and the underlying energy demands with MCM before and after healthy volunteers practiced a challenging visuo-spatial task for 4 weeks. Proceeding from the convergence of functional connectivity and glucose metabolism already during the first execution of a novel task[19] we expect that after continuous skill learning this task-specific association is consolidated also at resting-state. We hypothesize that training effects will be further reflected in the interaction of higher-order brain regions involved in cognitive control.

## Results

Multimodal brain imaging data and behavioral performance were acquired for 41 healthy participants in a longitudinal study design (Fig. 1). Subjects were assigned to either the training ($n = 21$) or control group ($n = 20$), which were carefully matched regarding the distributions of age, sex, and general intelligence (all $p > 0.5$). All subjects completed two PET/MRI scans ~4 weeks apart ($p = 0.2$ between groups) with simultaneous acquisition of fPET using [18F]FDG, functional connectivity (blood oxygen level-dependent signal (BOLD)) as well as fMRI (BOLD and arterial spin labeling (ASL), Supplementary Fig. S1). Imaging was obtained at rest and during the performance of a cognitively challenging task (an adapted version of the video game Tetris®) at two levels of difficulty (easy, hard), which required rapid visuo-spatial processing, motor coordination, and planning. Between the two PET/MRI measurements the training group practiced the cognitive task on a regular basis with the explicit aim to be able to manage the hard level at the second scan. Both PET/MRI sessions were accompanied by cognitive assessment to relate improvements in task performance to mental rotation, visual search, or spatial planning.

First, we combined the imaging parameters of glucose metabolism (CMRGlu), blood flow (CBF), and the BOLD signal for a functional delineation of brain regions with increased metabolic demands during task performance. In the next step, learning-induced changes in the networks encompassing the task-specific regions were investigated in the framework of MCM. MCM represents the associations between regional patterns of CMRGlu and BOLD-derived functional connectivity (FC), thereby enabling inference on directional connectivity[9,19]. For a thorough assessment of neuroplastic network effects, MCM was computed in an unbiased whole-brain approach for all three conditions (rest, easy, and hard task). In addition, simulations were carried out to identify whether the metabolic underpinnings or network reorganizations drive the training-induced changes by manipulating spatial MCM correlations based on values of CMRGlu of FC. Finally, changes in gray matter volume and white matter microstructure were assessed to identify whether learning-induced adaptations of MCM are also mirrored by structural adaptations. Please see the supplement for a thorough methodological description.

**Learning-induced improvement in cognitive skill performance.** Task performance was given by score per minute achieved during Tetris®, which underlined the successful training (Fig. 2). At the first PET/MRI measurement (i.e., before the training period), the training and control group did not differ regarding task performance, neither for the easy nor the hard task condition (both $p > 0.5$). Subjects in the training group then practiced $53.6 \pm 5.2$ min per day for $20.9 \pm 1.5$ days within a 4-week period using an online training program. This training period elicited significant changes in task performance (group*time*condition interaction, $p < 10^{-5}$),

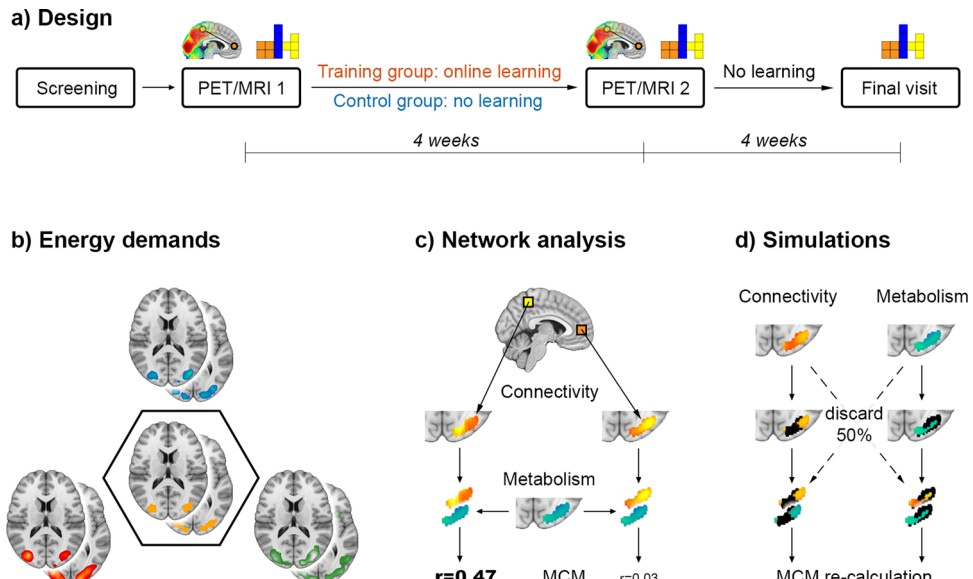

**Fig. 1 Design and analysis. a** After the initial screening, participants were randomly assigned to the training or the control group. All subjects underwent two simultaneous PET/MRI examinations for acquisition of structural, functional and metabolic data at resting-state and while performing a challenging visuo-spatial processing task (the video game Tetris®, Supplementary Fig. S1). In the 4-week period between the two PET/MRI scans, the training group regularly practiced the task using an online platform, whereas the control group did not. After the second PET/MRI scan, no further training was carried out and the training group completed a final task session on a laptop. Additional testing of different cognitive domains was performed at both PET/MRI examinations. **b** To obtain a robust estimate of task-specific increases in energy demands, the imaging parameters of glucose metabolism (blue), cerebral blood flow (green) and BOLD-derived activation (red) were combined in a conjunction analysis (intersection, orange). Joint active areas served as target regions for the subsequent network analysis (Supplementary Fig. S2). **c** We extended metabolic connectivity mapping (MCM) to the whole-brain level to assess learning-induced adaptations in directional connectivity towards regions with high task-specific energy demands. The BOLD signal of each brain voxel (exemplarily shown as yellow/orange squares) yields a certain functional connectivity pattern in the target region (here the occipital cortex). Computing the spatial correlation between patterns of functional connectivity (yellow/orange) and glucose metabolism (blue/green) results in an MCM value for each brain voxel that reflects the directional connectivity to the target. **d** Finally, simulations were carried out to disentangle the individual contribution of glucose metabolism and functional connectivity to MCM learning effects. Voxels in the target region were gradually removed based on values of connectivity or metabolism (here 50% black voxels in left and right columns, respectively), followed by recalculation of MCM values and the corresponding learning effects.

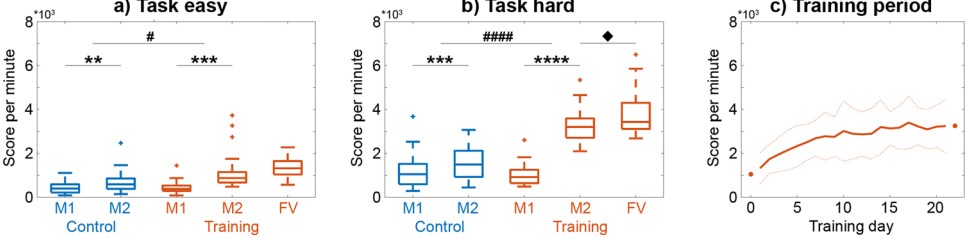

**Fig. 2 Behavioral data for the video game Tetris® measured as score per minute.** Changes in task performance differed between the two PET/MRI measurements (M1 and M2), groups and task conditions (group*time*condition interaction, $p < 10^{-5}$). **a** For the easy task condition, the training group ($n = 21$) showed a 2.7-fold increase in performance, which was significantly higher compared to the control group ($n = 20$). **b** For the hard task, changes in performance followed a similar pattern but effects were more pronounced, with the training group showing a 3.1-fold improvement in performance. Also, task performance for the hard task condition further increased even without training until the final visit (FV). The time between measurements/visits was 4 weeks. Initial performance at measurement 1 was not significantly different between the groups for both task conditions ($p > 0.5$). **c** Monitoring the task performance during the training period highlights the continuous improvement. The learning curve further matched with the performance of the two PET/MRI measurements as indicated by the dots (average values of M1 and M2 in **b**). Solid and dotted lines represent mean and standard deviation, respectively. Data were cut after 21 days as less than 1/3 of the subjects trained longer than this period. For **a** and **b**, post-hoc comparisons indicate significant differences for the group*time interactions ($^{\#}p < 0.05$, $^{\#\#\#\#}p < 10^{-9}$), for the differences between the two measurements ($^{**}p < 0.01$, $^{***}p < 0.001$, $^{****}p < 10^{-10}$) and for the difference between measurement 2 and the final visit ($^{\blacklozenge}p < 0.05$). All p-values were corrected for multiple comparisons with the Bonferroni-Holm procedure. Boxplots indicate median values (center line), upper and lower quartiles (box limits) and 1.5× interquartile range (whiskers). Data for the plots are provided in Supplementary Data 1–3.

with the training group performing significantly better in both task conditions after training. However, the improvement in task performance was particularly more pronounced for the hard task condition (group*time interaction, $p < 10^{-9}$, Fig. 2b) as compared to the easy condition ($p < 0.05$, Fig. 2a), indicating a more robust

learning effect in conjunction with higher task load and skill demand. In the training group, the task performance of the two PET/MRI scans also matched that obtained during the online training (Fig. 2c). Interestingly, task performance even further improved in the training group for the hard task level another

4 weeks after the second PET/MRI scan ($p < 0.05$, Fig. 2b), although no additional training was carried out. Furthermore, the training group ($p < 0.001$) but not the control group ($p > 0.1$) showed improved mental rotation performance after learning (group*time interaction $p < 0.05$). A similar pattern was observed for the visual search task (interaction $p = 0.05$, training group before vs. after learning $p = 0.09$, control group $p > 0.2$), but not for spatial planning performance (all $p > 0.6$).

**Overlapping task-specific neuronal activation across imaging modalities.** To identify regional increases in neuronal activation elicited by task execution, we combined three imaging parameters that represent different indices of metabolic demands obtained during the first PET/MRI measurement. CMRGlu, CBF, and BOLD signal changes during task execution (all $p < 0.05$ FWE-corrected) showed high spatial overlap in task-specific increases (Dice coefficient = 0.48–0.57, Supplementary Fig. S2), similar to our previous work[19]. Brain regions with mutual task-specific effects across imaging modalities (i.e., intersection) comprised the occipital cortex (Occ, 14.0 cm$^3$), intraparietal sulcus (IPS, 20.3 cm$^3$), and frontal eye field (FEF, 6.9 cm$^3$), with the latter two representing the dorsal attention network (DAN). These three areas served as target regions for the subsequent assessment of learning-induced changes with MCM. Two other clusters of overlapping task increases were observed in the ventral premotor cortex and occipital/temporal inferior cortex, which were however not further considered due to their limited spatial extent (0.3 and 0.9 cm$^3$, respectively).

**Learning-induced adaptations in metabolic connectivity mapping.** Proceeding from the above conjunction of metabolic demands during acute task execution, we investigated the corresponding network changes after practicing the same task over a 4-week training period by computing the association between CMRGlu and BOLD-derived FC. With Occ as the target region, learning-induced effects in MCM were observed in the dorsal anterior cingulate cortex (dACC) and the insula, both being integral parts of the salience network (SN, group*time*condition interaction, all $p < 0.05$ FWE-corrected, Fig. 3). This result was obtained independent of the contrast of interest, whereas two other regions (left insula and primary visual cortex) were observed just for one of the contrasts and will thus not be considered further. Post-hoc analysis showed that after the learning period MCM values of these two connections directed towards Occ increased at the resting-state for the training group as compared to the control group (group*time interaction, all $p < 0.01$). In contrast, MCM of connections from dACC and insula decreased during the execution of the hard task level ($p < 0.01$–0.05). Further analysis confirmed that these differences emerged exclusively from changes in the training group ($p < 0.01$–0.05) with no significant differences in the control group (all $p > 0.09$). MCM values obtained at the first PET/MRI measurement were not significantly different for the two mentioned connections. Moreover, the results remained stable when defining the target region only from task-specific CMRGlu and BOLD changes (i.e., without CBF, Supplementary Fig. S3). Furthermore, no significant training-induced effects in MCM were observed for the other two target regions that showed overlapping task-specific activation (i.e., FEF and IPS).

Based on the diverging training effects in the salience network (i.e., increased MCM at rest after training vs. decreased MCM during the task) we further tested whether the combination of these changes was related to cognitive performance. Indeed, the difference in MCM values between rest and the hard task for the connection from dACC to Occ was positively associated with the

Tetris® score of the second PET/MRI scan ($\rho = 0.46$, $p < 0.05$) and the area under the curve of scoring obtained during the entire 4-week training period ($\rho = 0.56$, $p < 0.01$, Fig. 4a–b). MCM values of the dACC were also associated with mental rotation performance obtained during cognitive testing (normalized duration, $\rho = -0.56$, $p < 0.01$, Fig. 4c).

**Differential role of glucose metabolism and functional connectivity in neuroplasticity.** In a simulation analysis, we disentangled the individual contributions of CMRGlu and FC to the training-induced changes in MCM described above (Fig. 5). This revealed that learning-specific increases in MCM at resting-state were dependent on CMRGlu, but not FC for the two connections (insula and dACC towards Occ). Conversely, the inverse pattern was observed for the hard task condition, where MCM training effects were driven by FC, but were largely independent of CMRGlu for both connections. However, random removal of voxels up to 90% did not affect the learning-induced MCM changes. This further supports the specificity of CMRGlu and FC driving MCM training changes and indicates that effects were not dependent on the size of the target region.

**Structural imaging.** There were no significant learning effects regarding gray matter volume or white matter microstructure (group*time interaction, all $p > 0.05$ FWE-corrected).

## Discussion

We employed brain network analyses of simultaneous PET/MR imaging to investigate learning-induced neuroplastic changes in functional network reorganization and the underlying metabolic demands that relate to cognitive performance improvements. MCM served as a suitable multimodal approach to assess task-specific and resting-state adaptations, which earlier have been examined independently. Four-week training of a visuo-spatial processing task resulted in adaptations of MCM from the salience network to the occipital cortex, with an increased association between glucose metabolism and BOLD-derived functional connectivity at resting-state, but decreases during the execution of the hard task condition. This divergence between resting-state and task-related MCM adaptations also explained cognitive performance after learning but was not simply driven by gray or white matter changes. Simulations further enabled a specific attribution of training effects at resting-state to CMRGlu and those during task execution to FC. Together, these findings highlight that the interaction of both metabolically expensive general neuroplastic adaptations and task-specific network reorganizations is required for improvement in the behavioral performance of visuo-spatial skill learning.

Similar to our previous work[19], task-specific activations were observed for the DAN and Occ. This result complies with the employed task as Occ is involved in visual attention-drawing[20,21] and provides spatial representations of the visual field[22,23], whereas the DAN is known for its role in controlling visuo-spatial attention[24]. However, the DAN represented by FEF and IPS showed no relevant training effects, indicating that this network mirrors a more general involvement in visuo-spatial processing required for planning and problem-solving of the current task.

In comparison, the main training effect was a pronounced alteration of the influence from the SN (insula and dACC) to Occ. The right anterior insula mediates switching between task-irrelevant and task-specific networks that convey externally oriented attention[25–28] through the dorsal visual pathway and the intraparietal sulcus[26,29]. This pathway thus represents a crucial bottom-up connection to the SN with the anterior insula as a feedforward stimuli filter and working memory access hub[26,30,31].

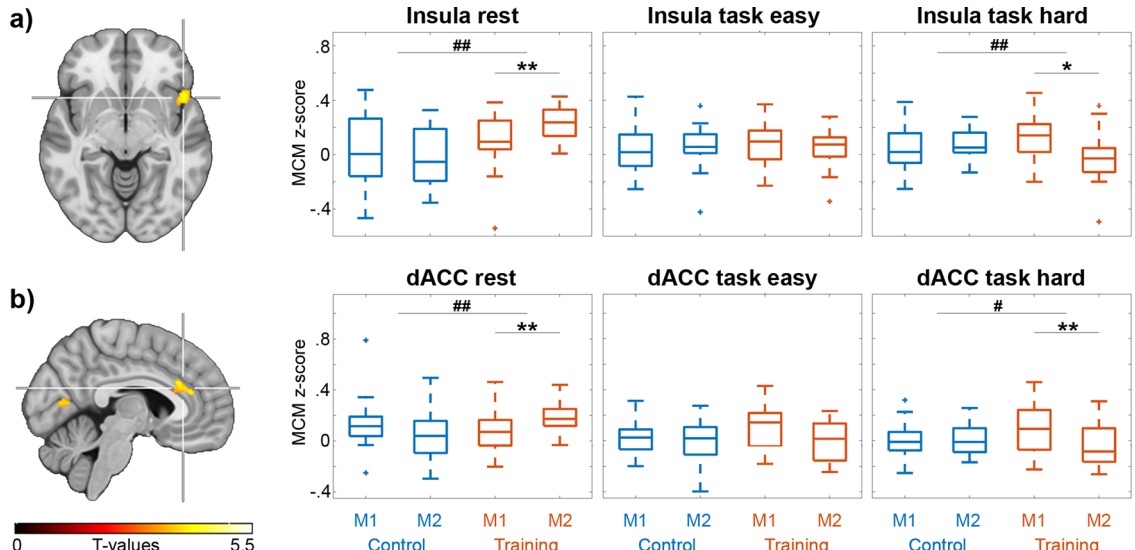

**Fig. 3 Learning-induced changes in metabolic connectivity mapping (MCM) with the occipital cortex as target region.** Four weeks of training the video game Tetris® resulted in specific adaptations of connectivity from the right insula (**a**) and the dorsal anterior cingulate cortex (dACC, **b**) to the occipital cortex (group*time*condition interaction, $p < 0.05$ FWE-corrected cluster level). Post-hoc comparisons showed that at rest MCM increased for both connections in the training group ($n = 21$) as compared to the control group ($n = 20$). In contrast, MCM decreased during the hard task condition in the training group. There were no significant changes in the control group between the two measurements (M1, M2). Furthermore, MCM values between training and control groups at measurement 1 were not significantly different. Boxplots show the MCM z-scores of the clusters indicated by the crosshair. Post-hoc comparisons indicate significant differences for the group*time interaction (#$p < 0.05$, ##$p < 0.01$) and for the differences between the two measurements (*$p < 0.05$, **$p < 0.01$), corrected for multiple comparisons with the Bonferroni-Holm procedure. Boxplots indicate median values (center line), upper and lower quartiles (box limits) and 1.5× interquartile range (whiskers). Data for the plots are provided in Supplementary Data 4–7.

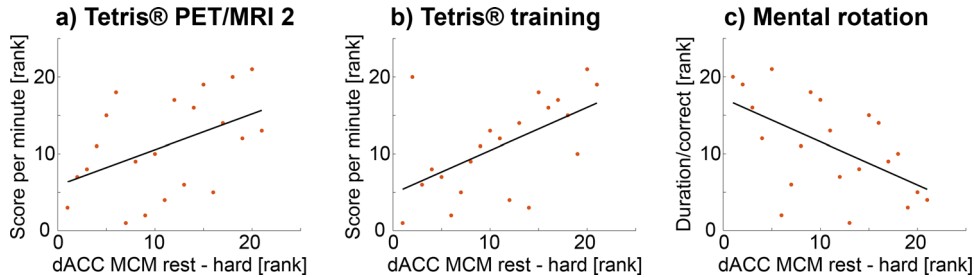

**Fig. 4 Associations between MCM adaptations and cognitive performance.** Based on the training-induced effects in the salience network (Fig. 3), one would expect that subjects with high MCM values at rest and low values during task execution (i.e., a high divergence between rest and task) show the best cognitive performance after learning. Thus, the difference of MCM values between rest and the hard task condition at the second PET/MRI scan was correlated with task performance. Positive associations of dACC MCM values were observed with the Tetris® score (high score = high performance) of the second PET/MRI measurement (**a**, rho = 0.46, $p < 0.05$) and that obtained during the 4-week training period (**b**, rho = 0.56, $p < 0.01$, normalized area under curve). Further, dACC MCM values were negatively associated with the mental rotation performance (duration/number of correct answers with low value = high performance, **c**, rho = −0.56, $p < 0.01$). All values were rank transformed to account for one outlier, thus correlation values represent Spearman's rho ($n = 21$). Data for the plots are provided in Supplementary Data 8–10.

On the other hand, the dACC plays an essential role in cognitive control[32–37] error monitoring[38–40] and negative feedback[41], whereas inhibition of dACC functioning results in poorer accuracy and reduced learning[42]. These cognitive aspects have led to an integrative account of dACC function to provide optimized task representations through adaptive coding of task-relevant variables, which further guide behavior[39].

Furthermore, the anterior insula and dACC were both identified to exert activation patterns preceding task errors[43] and were associated with performance monitoring[35]. Since training led to a marked improvement in task performance, the associated decrease of the insula and dACC input to Occ during task execution might indicate a reduced error rate. The observation of changes in cognitive performance and MCM mainly for the hard task condition is well in line with other reports[44,45] and the

experimental design of our Tetris® task, where only the hard level was set to require specific training. The accompanying behavioral data further suggest that improvements in task performance were specific to abilities of mental rotation and, to a lesser extent, visual search and working memory, but not spatial planning and problem-solving. Following the assumption that mental rotation reflects a continuous transformation performed on visual representations in the human brain[46], we speculate that the observed improvement is based on the representational adaptation described below.

To sum up, the observed adaptations of SN connections indicate an optimized hierarchical top-down influence on the Occ. In this context, the anterior insula seems to play a unique role as the switching point between bottom-up saliency and top-down control, whereas the dACC provides monitoring of

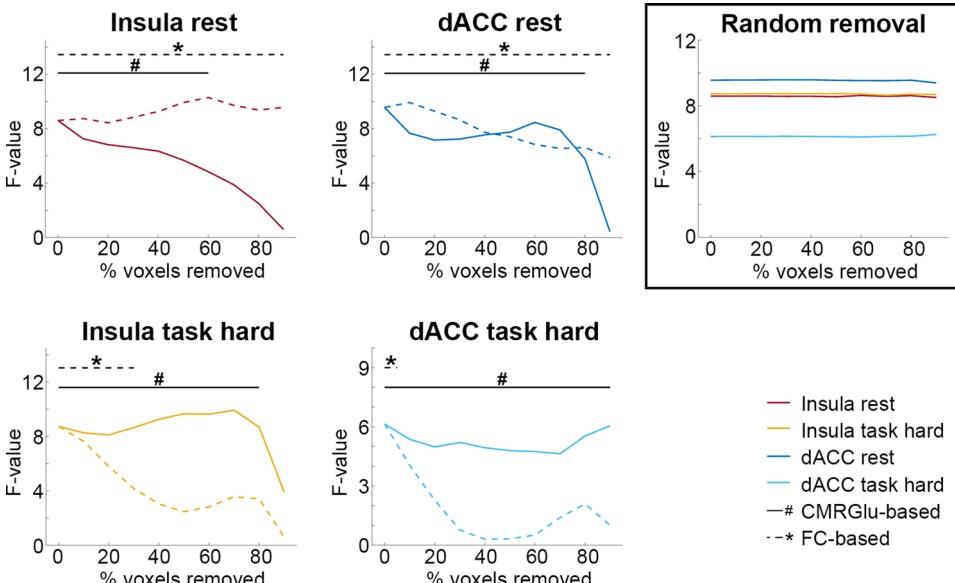

**Fig. 5 Simulated perturbations of learning-induced changes in MCM.** We aimed to identify whether learning-specific MCM effects in Fig. 3 were driven by glucose metabolism (CMRGlu) or functional connectivity (FC). Voxels of the occipital cortex (i.e., the MCM target region) were progressively removed based on increasing values of CMRGlu (solid lines) or FC (dashed lines) and training effects were recalculated (F-value of group*time interaction). At resting-state simulated removal of voxels based on CMRGlu abolished training-induced MCM effects for both connections towards the occipital cortex, which was, however, not the case for FC (top row). The inverse pattern was observed for the hard task condition, where training-specific decreases in MCM were nullified when removing voxels based on FC, but not CMRGlu (bottom row). #/solid black lines: $p < 0.05$ when removing voxels based on CMRGlu. */dashed black lines: $p < 0.05$ when removing voxels based on FC values. 0% of voxels removed represents the results shown in Fig. 3 (i.e., when using the entire target region). Of note, randomly removing up to 90% of voxels in the occipital cortex did not affect the learning-induced changes in MCM at all (right top panel, all $p < 0.05$), highlighting the specificity of CMRGlu and FC to drive MCM changes and indicating that effects are not dependent on the size of the target region. The colors for the random removal match those of the other panels. Data for the plots are provided in supplementary data 11–13.

cognitive control and adjustment of task representations. These mechanisms emphasize a successive shift from basic task execution, instantly implemented by feedforward activation through Occ to the DAN[19], to the optimized top-down control of salient input from the SN to Occ after successful training, thus enabling improved cognitive performance.

The hierarchical interaction across brain regions as assessed with MCM revealed divergent yet complementary effects of the learning process at resting-state and during task execution. This approach further enabled us to provide a unified interpretation of skill learning as an advancement of regionally specific neuronal representations of the task.

While the retinal image is represented by early visual pathway areas, higher-order brain regions create edited representations of the visual field to fulfil immediate goals of attention and behavior as a function of task demands[47]. Our findings thus integrate the spatio-temporal dynamics underlying bottom-up and top-down attentional control via the SN[26] into the conception of learning to reduce prediction errors between task representations of brain regions of different processing levels, based on synaptic modifications[48].

The initial learning stage comprises interactions with unknown task demands and high perceptual load, thus requiring adaptation of attention due to limited processing resources[49,50]. Attention can be understood as a Bayesian optimization of hierarchical perception to infer the precision of a probabilistic representation of the environment[48,51]. The model is also supported by rodent studies, where projections from the ACC to the visual cortex encoded the discrepancy between predicted and actual sensory input[52–55]. The attentional optimization is implemented across different hierarchical levels of cortical systems[56,57], based on synaptic gain or responsiveness of

postsynaptic neurons that encode prediction errors[51,58]. Here, so-called state and error units characterize the different representations and their inaccuracy (i.e., the prediction error), respectively[48]. The prediction error itself is optimized through top-down modulatory control[48], e.g., by the SN[26], which enables to select[49,59,60] and sharpen[50] representations and to distinguish between efficient and unsuccessful task rules, thereby implementing an advanced representation of the task[50,61].

Training will continuously alter the task representation by incorporating relevant and discarding irrelevant information, approaching an optimal solution, and finally yielding a sparse representation[62,63]. Following a sufficiently long training period, the task representation is saved within these state units. We refer to this stored representation as 'skill engram', extending the term 'engram', first introduced in 1904 to describe memory representations[64]. For the herein employed task the skill engram is equivalent to a specific pattern of functional connectivity that matches the underlying metabolic demands between SN and Occ as reflected by MCM increases at the resting-state. Importantly, the stored skill engram can be retrieved instantaneously at the next task performance. Then, the optimized representations yield a small prediction error between higher- and lower-order brain regions. A minimized prediction error requires only minor cognitive control, resulting in decreased directional influence (lower MCM values) from SN to Occ during task execution. We would like to note that these representations can still be subject to adaptations and reconsolidation with further training, highlighting that engrams are dynamic in nature[65–67]. This strongly emphasizes that the observed effects in resting-state and during task execution complement each other by representing the stable and fluid/plastic components of memory (or skill) traces, respectively[66]. The above interpretation is further supported by

our observation that a high divergence of neuroplastic changes between rest and task was associated with improved task performance.

In this study, the complementary task- and resting-state learning effects were identified by simultaneous PET/MR imaging. While interpreting our results at the neurobiological level, we acknowledge that the following section relies on previous work mostly obtained from preclinical studies. We propose that the dynamic process of representational advancement is molecularly implemented by synaptic tagging and capture, together referred to as long-term potentiation[68–70], whereas the skill engram's storage and retrieval might comply with synaptic (re) consolidation.

Neuronal stimulation through task performance elicits synaptic tagging, which describes short-lasting adaptations of the postsynaptic density of dendritic spines. We interpret a high number of tagging events as high prediction error, since numerous different traces are explored, which is reflected in high task-specific MCM before the learning process. Repeated tagging through continuous training of the task then induces synaptic capture, where plasticity-related products (e.g., ARC, AMPA receptor subunit GluR1) are mobilized to stabilize the spine architecture. These structural adjustments enable functional synaptic potentiation by anchoring additional postsynaptic AMPA receptors[71], finally representing the consolidated skill engram. Importantly, the synaptic adaptations of the learning process beyond tagging are metabolically expensive[72]. Particularly the insertion of AMPA receptors has been shown to double the postsynaptic energy consumption in terms of ATP[11]. Since BOLD-derived FC reflects glutamate-mediated processes[73], we propose that the observed MCM increases at rest (i.e., the increased association between FC and CMRGlu) indicate long-term potentiation via an increased expression of glutamatergic AMPA receptors. As mentioned above, this consolidated engram obtained through training results in a low prediction error during task execution and thus also in a few additional tagging events, which is reflected in a decreased task-specific MCM value after training (summarized in Fig. 6).

Recent metabolic simulations of synaptic adaptations supported the neurobiological process of synaptic tagging and capture as a plausible physiological mechanism to increase energy efficiency[66,74]. Computations emphasized that storage of transient memories without protein synthesis is up to ten-fold more energy-efficient, compared to the immediate formation of long-term potentiation that requires a high amount of metabolic resources[74]. This is in line with our own simulations, indicating that energy-demanding (synaptic capture) learning effects at rest were dependent on CMRGlu, while transient storage of information (synaptic tagging) elicited by task execution was driven by FC. We therefore speculate that simulations affecting CMRGlu equal a perturbation of state units that represent the skill engram at resting-state. On the other hand, FC-based deletion may disrupt the ability to establish a correct prediction error during task execution as encoded by error units. Thus, the two aspects will either alter the molecular basis of neuronal task representations or the actual cognitive control, respectively. However, disruption of any of the two will nullify the training effect, again highlighting the complementary importance of both effects for successful learning.

We are aware that causal relationships between the observed findings from human brain imaging and more invasive work underpinning the prediction error model as well as the process of synaptic tagging and capture still need to be established. Although these models can be unified in our data, future work should aim to resolve this knowledge gap by directly testing the suggested associations.

In sum, applying MCM to simultaneous PET/MR imaging data of visuo-spatial learning enabled the combined assessment of brain network dynamics and their underlying metabolic demands at resting-state and during task performance. Future studies may build on these findings by investigating if the described effects can be transferred to other learning paradigms or if they are specific to the visuo-spatial domain. Although we have carefully assessed the cognitive domains relevant for the Tetris® task, we acknowledge the possibility that further aspects may have an effect when investigating brain function longitudinally. However, testing these would have exceeded the scope of this work.

Another limitation of our work is that only two time points of the learning process were assessed (baseline and 4-week training). In this context, future investigations should determine the minimum training time to obtain an effective skill engram and how long it remains stable after the training is terminated. Our behavioral data suggest that efficient task execution persists for at least after 4 weeks without further training. Finally, it needs to be determined if vulnerable development phases or adverse life events may alter the learning process. Disentangling the metabolic and functional requirements for neuroplasticity might prove beneficial to differentiate between different forms of neurodegenerative diseases and evaluate the severity of tissue damage in traumatic brain injury, feasibly with the use of cohort-specific or individually adapted tasks. Furthermore, the approach might prove valuable for assessing cognitive deficits in psychiatric disorders like depression or schizophrenia and their respective treatment. Thus, the application in patient cohorts may identify if pathological deficits in learning and memory are driven by the failure to establish the engram on a molecular basis or insufficient functional error optimization.

## Methods
The cognitive task as well as PET/MRI data acquisition and first-level analyses have been described in detail in our previous cross-sectional work[19].

**Experimental design**. In this longitudinal study participants were randomly assigned to the training or control group (dynamic balanced randomization stratified by age, sex, and general intelligence) and underwent two PET/MRI measurements (training: 28.0 ± 1.2 days apart, control: 29.5 ± 5.0 days, Fig. 1). The training group practiced the cognitive task regularly between the two imaging sessions while the control group did not perform a task. After the second PET/MRI scan, also the training group stopped practicing the cognitive task but completed the last task session on a laptop at the final visit to assess long-term skill consolidation (30.6 ± 2.8 days after the second PET/MRI scan).

During each PET/MRI measurement subjects performed a challenging cognitive task at two predefined levels of difficulty (easy, hard). The PET/MRI acquisition lasted 100 min and has been described in detail in the previous work[19]. Briefly, this included a T1-weighted structural (8 min) and diffusion-weighted scan (12 min) as well as blood oxygen level-dependent (BOLD), arterial spin labeling (ASL), and simultaneous fPET imaging (Supplementary Fig. S1). Prior to the administration of the radiotracer [18F]FDG, BOLD and ASL was acquired at rest. Subjects were instructed to look at a crosshair, relax and not focus on anything in particular. Afterwards, fPET started with an initial baseline (8.17 min) and subsequent periods of continuous task performance (6 min for two easy and two hard conditions, randomized). During the task, BOLD and ASL data were recorded in pseudorandomized order. BOLD data acquired at rest and during continuous task performance was used for the computation of metabolic connectivity mapping. All periods of task execution were followed by periods of rest (5 min). After fPET was completed, BOLD data were also acquired in a conventional block design (four 30 s blocks of easy, hard, and control conditions each, randomized with 10 s baseline in-between, 8.17 min) to obtain another proxy of task-specific activation.

**Cognitive tasks and online training**. The cognitive task comprised an adapted version of the video game Tetris®[19] (https://github.com/jakesgordon/javascript-tetris, MIT license), implemented in electron 1.3.14, and was chosen for various reasons. First, a computerized task enables execution during the training period and also within the PET/MRI scanner (as opposed to e.g., juggling[75,76]), offering the key possibility to assess learning effects at rest and during task performance. Also, the individual task performance can be easily monitored to follow the training success of the participants. Furthermore, the task is well suited to maintain high levels of attention during prolonged and continuous task performance, which in turn provides an optimal setting for fPET imaging[77]. Finally, the computerized task enables the programming of different levels of task difficulty. Thereby, we intentionally set the easy level to be manageable for a novice, whereas the hard level

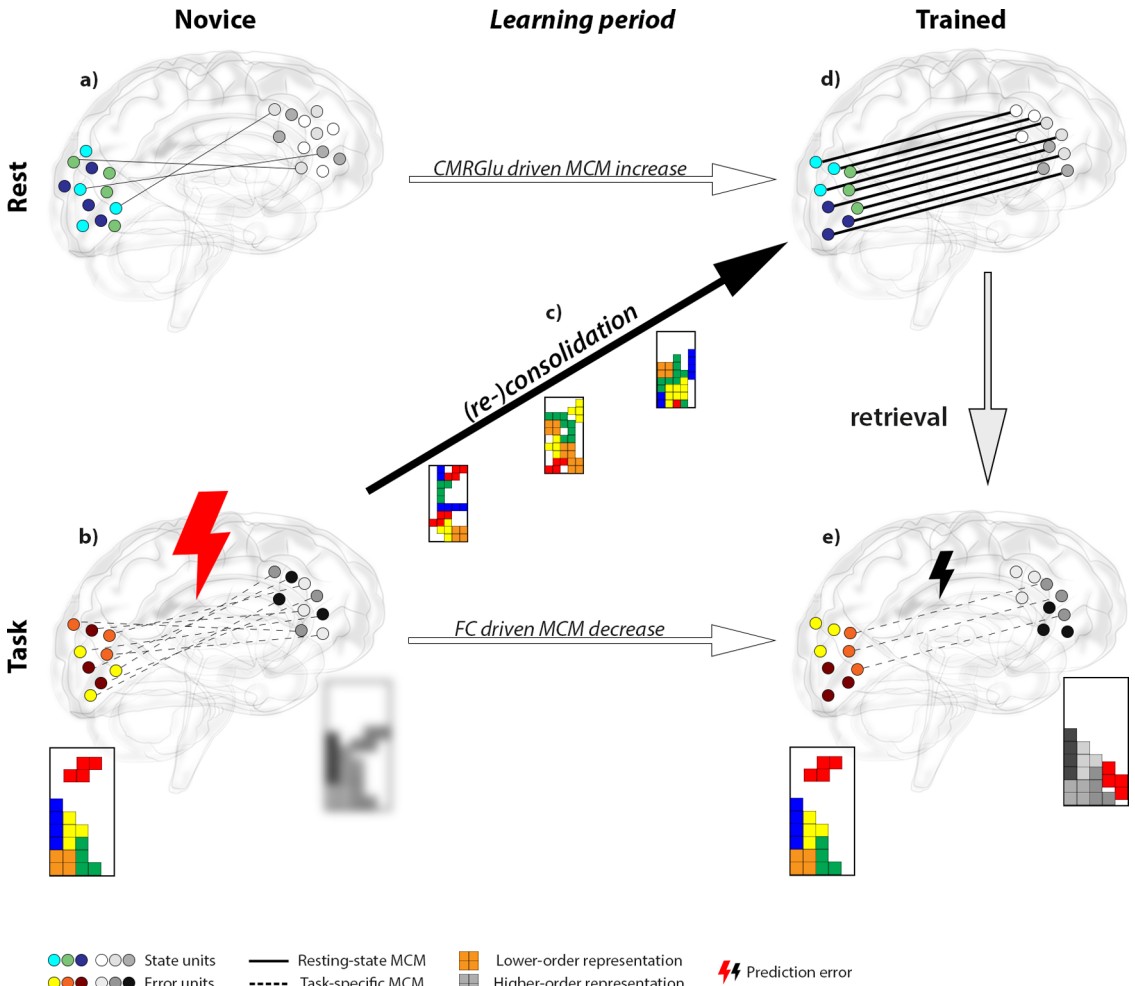

**Fig. 6 Schematic illustration of training effects and potential neurobiological mechanisms.** Cognitive skill learning results in complementary metabolic adaptations at rest (top row) and functional network reorganization during task execution (bottom row). Glass brains depict directional connectivity from the higher-order salience network (grayscale circles) to the lower-order occipital region (colored circles) as assessed with metabolic connectivity mapping (MCM, number of lines). **a** Training naïve subjects exhibit low directional connectivity at resting-state (solid lines) between unorganized state units (blue-green circles), since the skill trace is not yet established (random circle arrangement, crossing lines). **b** Due to the lack of training, higher-order representations in the SN are inaccurate (blurred grayscale Tetris®) in comparison to lower-order visual sensory information (colored Tetris®), resulting in a high prediction error (large thunderbolt) encoded by error units (red–yellow circles). The representational inaccuracy requires substantial dynamic optimization between brain regions of different hierarchies (numerous dashed lines). **c** With repeated task performance during the learning period functional network reorganization approaches an optimal solution. Presumably, this is realized by a high frequency of synaptic tagging, where optimal task representations are gradually encoded in the salience network by synaptic capture and subsequent anchoring of glutamatergic AMPA receptors. **d** After the learning period, state unit directional connectivity increases, which equals the consolidated skill engram (parallel lines between organized circles). The metabolic emphasis of this process suggests the energy-intensive formation of clustered and potentiated synapses (line thickness). **e** The established skill engram can then be retrieved for task execution. This results in a decreased prediction error (small thunderbolt) as representations between higher- and lower-order brain regions became more accurate (sharpened grayscale Tetris®). Thus, only minor cognitive control is required (few dashed lines) to apply an efficient task strategy. In sum, these observations indicate that effects of skill learning at resting-state and during task execution are two sides of the same coin, where different neurobiological mechanisms complement each other to improve task performance. The glass brain was kindly provided by Dr. Gill Brown (https://neuroscience-graphicdesign.com/) under CC BY-NC 4.0.

requires specific training. We therefore expected the most pronounced effects regarding behavioral data and neuroplasticity for the hard task condition. In this task bricks of different shapes drop from the top of the screen. By rotation and alignment of the bricks, the aim is to build complete horizontal lines which disappear and increase the score. Bricks were moved by operating four buttons with the right hand only (index finger: move brick left, middle: rotation, ring: down, small: right; same fixed assignment for in-scanner and online version).

For the PET/MRI measurements, the two levels of difficulty differed regarding the speed of the falling bricks (easy/hard: 1/3 lines per sec) and the number of incomplete lines that were already built at the bottom (easy/hard: 2/6 lines out of 20). An additional control condition was employed for the BOLD block design, where bricks were navigated through a chimney and disappeared at the bottom. Participants familiarized themselves with the task and control buttons by a 30 s training of each task condition right before the scan started. For the online learning

phase between the two PET/MRI scans participants were instructed to train on a regular basis (45–60 min per session, at least 20 sessions within 4 weeks). An individual username and password were provided to access the web-based training platform in the web browser Chrome. Subjects were able to select any combination of brick speed and prebuilt lines freely. However, the explicit aim of the 4-week training period was to be able to manage the hard task condition as carried out in the PET/MRI session. The implemented scoring was identical for the PET/MRI measurements and online training, following the original scheme of Tetris®. The score for each complete line was $k*(n+1)$, with n being the speed level of the falling bricks and k representing the score for one (40), two (100), three (300), or four (1200) lines removed simultaneously.

At each PET/MRI examination, participants also completed a cognitive testing session outside the scanner to relate improvements in task performance to specific cognitive domains of mental rotation, visual search, and working memory, as well

as spatial planning and problem-solving. For the mental rotation task, subjects viewed pairs of 3D objects (similar to Tetris bricks) and indicated if the objects are congruent after mental rotation or not[78] (images obtained from http://www.tarrlab.org/ under CC BY-NC-SA 3.0). The paradigm comprised 20 stimuli pairs of 40°, 80°, 120°, and 160° rotation (5 each) and 50% congruent pairs (random presentation order). Abilities of visual search and working memory were tested with the delayed match to sample task[79]. After the presentation of an abstract pattern ($10 \times 10$ square of black and white tiles) for 1 s and a delay of another second, subjects were required to choose the matching pattern from a set of 2, 4, 6, or 8 similar patterns (3 each, 12 stimuli sets in total, random presentation order). Spatial planning and problem-solving abilities were assessed with the Tower of London task[80,81]. Within the task, starting from an initial configuration of three colored discs on three pegs of different height, subjects are required to move the discs to reach a specific target configuration. The paradigm included 3–8 moves (2 each) and 12 stimulus sets in total (random presentation order).

**Participants.** For this study, 53 healthy subjects were initially recruited and data from 41 healthy subjects were included in the analysis (all right-handed). Reasons for study dropout were voluntary discontinuation ($n = 6$), omission to acquire fPET due to issues with arterial cannulation or radiotracer synthesis ($n = 3$), failure of arterial blood sampling ($n = 2$), and excessive head motion during the BOLD acquisition ($n = 1$). Among the 41 subjects, 21 were assigned to the training group (mean age ± sd = $23.0 \pm 3.6$ years, 11 women) and 20 to the control group ($23.1 \pm 3.1$ years, 10 women). For two subjects of the training group, no ASL data were available due to technical issues. As no longitudinal MCM studies are available, the sample size was based on previous cross-sectional work using this technique[9,19]. Parts of this sample were already included in previous studies for the assessment of cross-sectional data[19] and test-retest reliability[82]. At the screening visit, the general health of all subjects was ensured through a routine medical examination performed by an experienced psychiatrist, including blood tests, electrocardiography, neurological testing (comprising examinations of mental status, cranial nerves, motor system, deep tendon reflexes, sensation, cerebellum) and the structural clinical interview for DSM-IV. All subjects also underwent a shortened version of the Raven standard progressive matrices test at the screening visit as an index of general intelligence (training group: $113.3 \pm 9.5$, control group: $115.1 \pm 9.0$; two parameter logistic model[83]). Urine pregnancy tests were carried out for female participants at the screening visit and before each PET/MRI measurement. Exclusion criteria were current and previous somatic, neurological or psychiatric disorders (12 months), substance abuse or psychopharmacological medication (6 months), current pregnancy or breastfeeding, previous study-related radiation exposure (10 years), bodyweight of more than 100 kg for reasons of radiation protection, MRI contraindications and previous experience with the video game Tetris® within the last 3 years. Experience with and regular playing of similar video games, specifically games primarily involving visuo-spatial skills like "Candy Crush," was another explicit exclusion criterium. Furthermore, participants of both groups were instructed not to play and especially not to learn any (new) video games while participating in the study. After a detailed explanation of the study protocol, all participants provided written informed consent, and they were insured and, after study completion, reimbursed for participation. The study was approved by the ethics committee of the Medical University of Vienna (ethics number: 1479/2015) and procedures were carried out in accordance with the Declaration of Helsinki. The study was pre-registered at ClinicalTrials.gov (NCT03485066).

**PET/MRI data acquisition.** Participants had to fast for at least 5.5 h before the start of the PET/MRI scan (except for unsweetened water), according to [18F]FDG procedure guidelines[84]. The radiotracer 2-[18F]fluoro-2-deoxy-D-glucose ([18F] FDG) was administered in a bolus (510 kBq per kg per frame for 1 min) plus infusion protocol (40 kBq per kg per frame for 51 min) with a perfusion pump (Syramed μSP6000, Arcomed, Regensdorf, Switzerland) which was kept in an MRI-shield (UniQUE, Arcomed)[77].

MRI recordings included a structural T1-weighted acquisition (MPRAGE sequence, TE/TR = 4.21/2200 ms, TI = 900 ms, flip angle = 9°, matrix size = $240 \times 256$, 160 slices, voxel size = $1 \times 1 \times 1$ mm + 0.1 mm gap, 7.72 min) and diffusion-weighted imaging (EPI sequence, TE/TR = 86/8800 ms, 64 diffusion directions with b-value = 1000 s per mm², 6 b0-images, matrix size = $104 \times 104$, 70 slices, voxel size = $2 \times 2 \times 2$ mm, 11.73 min).

Simultaneously with fPET, functional MRI was obtained using ASL (2D pseudo-continuous ASL sequence, TE/TR = 12/4060 ms, post label delay = 1800 ms, flip angle = 90°, matrix size = $64 \times 64$, 20 slices, voxel size = $3.44 \times 3.44 \times 5$ mm + 1 mm gap, $3 \times 6$ min[85]) and BOLD imaging (EPI sequence, TE/TR = 30/2000 ms, flip angle = 90°, matrix size = $80 \times 80$, 34 slices, voxel size = $2.5 \times 2.5 \times 2.5$ mm + 0.825 mm gap, $3 \times 6$ min for functional connectivity and 8.17 min for neuronal activation in the block design).

**Blood sampling.** Blood glucose levels were assessed right before the PET/MRI scan (Glu$_{plasma}$, triplicate) and were $5.45 \pm 0.73$ mmol per l. Manual arterial blood samples were taken at 3, 4, 5, 14, 25, 36, and 47 min after starting the radiotracer administration, providing a sufficiently sampled input function for the used bolus plus infusion protocol[77]. For all samples, whole-blood activity and following centrifugation also plasma activity were measured in a gamma-counter (Wizard², Perkin Elmer). Whole-blood activities were linearly interpolated to match PET frames and multiplied with the average plasma-to-whole-blood ratio, yielding the arterial input function.

**Cerebral metabolic rate of glucose metabolism (CMRGlu).** The reconstruction and processing of fPET data were carried out as described previously[77]. PET list mode data were corrected for attenuation with an established database approach[86] and reconstructed to frames of 30 s (matrix size = $344 \times 344$, 127 slices). Preprocessing was done in SPM12 (https://www.fil.ion.ucl.ac.uk/spm/) and included motion correction (quality = 1, register to mean), spatial normalization to MNI space via the T1-weighted structural MRI, and spatial smoothing with an 8 mm Gaussian kernel. Masking was applied to include only gray matter voxels and a low pass filter was employed with the cutoff frequency being 3 min (i.e., half the task duration). The general linear model was used to separate baseline from task-specific effects. The four regressors described the baseline, both task conditions (easy and hard, linear ramp function with slope = 1 kBq per frame), and head motion (the first principal component of all six motion regressors). The baseline regressor was computed as the average time course of all gray matter voxels, excluding those activated during the hard task of the individual BOLD block design ($p < 0.05$ FWE-corrected at voxel level). This multimodal approach has been shown to provide the best model fit[77]. Furthermore, including the BOLD changes in the baseline definition does not affect fPET task effects[77] or their test-retest reliability[82]. For absolute quantification of glucose metabolism, the Patlak plot was applied to calculate the influx constant, K$_i$. This was subsequently converted to CMRGlu by

$$\text{CMRGlu} = K_i * \text{Glu}_{plasma}/\text{LC} * 100 \qquad (1)$$

with the lumped constant (LC) set to 0.89. This procedure yields separate maps of CMRGlu at baseline and for each task condition (easy, hard), which were then used as an index of neuronal activation and for computation of metabolic connectivity mapping (see below). The approach has been shown to yield excellent test-retest reliability for CMRGlu at rest and fair-to-good reliability during task performance[82].

**Cerebral blood flow (CBF).** ASL data were processed according to standard procedures[87]. Voxels with a signal intensity below 0.8 times the mean value were set to zero to remove spurious effects, followed by motion correction in SPM12 (quality = 1, register to mean). The equilibrium magnetization of the brain M$_0$ was calculated as the average of all non-labeled images. A brain mask was computed from the M$_0$ image with the brain extraction tool (FSL, https://fsl.fmrib.ox.ac.uk/fsl/fslwiki/). All data were masked accordingly, spatially normalized to MNI space via the T1-weighted MRI (as for the fPET data), and spatially smoothed with a 8 mm Gaussian kernel. CBF was then calculated by

$$\text{CBF} = \frac{\lambda \Delta MR_{1a}}{2\alpha M_0 \{\exp(-\omega R_{1a}) - \exp[-(\tau + \omega)R_{1a}]\}} \qquad (2)$$

In this equation $\lambda = 0.9$ ml/g represents the blood-tissue water partition coefficient, $\Delta M$ the pairwise difference between labeled and non-labeled images, $R_{1a} = 0.67$ s⁻¹ the longitudinal relaxation rate of blood, $\alpha = 0.8$ the tagging efficiency, $\omega$ the post-labeling delay adapted for slice timing effects (=1800 at slice 1) and $\tau = 1508$ ms the duration of the labeling pulse. CBF was then averaged across time series separately for each condition (rest, easy, hard). As the maps obtained during task performance represent the sum of resting and task effects, the CBF at rest was subtracted to finally obtain the sole task-specific changes in CBF as a further proxy of neuronal activation.

**Blood oxygen level-dependent (BOLD) signal changes.** BOLD data of the block design were processed with SPM12 as described previously[77]. Data were corrected for slice timing effects (reference = middle slice) and head motion (quality = 1, register to mean). Similar to fPET and ASL, normalization to MNI space was carried out via the T1-weighted MRI, followed by spatial smoothing with an 8 mm Gaussian kernel. Task-specific changes were estimated with the general linear model, which included one regressor for each task condition (easy, hard, and control) and nuisance regressors for motion, white matter, and cerebrospinal fluid. The contrasts of interest used for the subsequent analyses were easy vs. control and hard vs. control.

**Overlap in task-specific neuronal activation.** The three different indices of task-specific metabolic demands (CMRGlu, CBF, BOLD) were combined to obtain a robust estimate of regions involved in task processing. This approach was chosen to enable comparison to our previous work[19] and to maximize the specificity of the MCM target regions. Still, we also compute the main MCM training effects for the combination of CMRGlu and BOLD only (Supplementary Fig. S3). For each imaging modality, separate one-sample t-tests were performed for the hard task condition of the first PET/MRI measurement across the entire sample. As ASL data were missing for two subjects, the sample used for this overlap analysis was $n = 39$. The resulting statistical maps were thresholded ($p < 0.05$ FWE-corrected voxel level), binarized, and combined in a conjunction analysis by computing the intersection across all three imaging modalities (Supplementary Fig. S2). The brain regions identified in this analysis were used as target regions for the subsequent MCM analysis, where homologous regions in the left and right hemispheres were

combined. The spatial overlap between the different imaging modalities was assessed by the Dice coefficient applied to the thresholded and binarized maps.

**Metabolic connectivity mapping (MCM)**. MCM is a multimodal framework that investigates the association of regional patterns between glucose metabolism and BOLD-derived functional connectivity (FC)[9]. Considering that the majority of energy demands emerge post-synaptically[10–12], the incorporation of CMRGlu identifies the target region and thus the MCM framework allows to assign directionality to a specific connection. In short, it is assumed that the seed region exerts an influence on the target region, yielding a particular voxel-wise FC pattern. If this influence is indeed causal, it will result in a corresponding CMRGlu pattern due to the coupling between the BOLD signal and metabolism[13,14], which is reflected in a non-zero MCM value given by the spatial correlation between patterns of CMRGlu and FC. For a detailed description of the underlying neurobiological effects of this approach and the analysis details the reader is referred to the previous work[9,19].

FC was computed at rest and for the two task conditions (easy, hard) from continuously acquired BOLD data using the preprocessing pipeline employed for the BOLD block design described above. After spatial smoothing, motion scrubbing was carried out to remove spurious connectivity induced by head motion[88]. Rotational motion parameters were converted to mm (displacement on a 50 mm sphere) and summed with the translational parameters. Frames with a displacement > 0.5 mm (plus one frame back and two forward) were discarded resulting in an average removal of $4.3 \pm 4.7\%$ of frames. There was no significant difference in head motion between groups, PET/MRI measurements, and conditions ($p > 0.3$ for all interactions and post-hoc tests). Subsequently, further confounding signals were removed by linear regression (motion parameters, white matter, cerebrospinal fluid) and a bandpass filter ($0.01 < f < 0.15$ Hz, enabling comparison of connectivity at rest and task performance[89]).

In this work, we extended the MCM framework from a region-of-interest to the whole-brain level, thereby avoiding a bias inherent to the a priori selection of brain regions. First, a specific target region B was chosen, herein defined as the overlap of neuronal activation across imaging modalities (see results and Supplementary Fig. S2). Next, the BOLD signal of any single voxel $A_i$ in the brain was (temporally) correlated with the time course of all voxels in the target region B and z-transformed, yielding a voxel-wise pattern of FC in B. This FC pattern of B was then (spatially) correlated with the corresponding CMRGlu pattern of B. The resulting MCM value was again z-transformed and assigned to the voxel $A_i$. Repeating the computation for every voxel $A_i$ in the brain gives a whole-brain map of MCM, where each voxel $A_i$ represents the directional connectivity to the target region B.

Previous studies have demonstrated that MCM is not affected by the size of the target region[9], spatial smoothing, or the preprocessing order of FC data[19]. Furthermore, directionality inferred from MCM has been validated by dynamic causal modeling[19].

**Gray matter volume**. T1-weighted structural images were segmented and spatially normalized to MNI-space using the longitudinal pipeline implemented in the CAT12 toolbox for SPM12 with default parameters to detect learning-induced changes. Gray matter volume was calculated by multiplication of the corresponding segment with the Jacobian determinants, adjusting for nonlinear deformation effects. The resulting images were spatially smoothed with an 8 mm Gaussian kernel.

**White matter microstructure**. Diffusion-weighted images were processed with FSL as described previously[90]. This included removal of non-brain tissue with the brain extraction tool as well as correction for eddy currents, head movement, and distortions with outlier replacement. For estimation of the diffusion tensors, the rotated b-vectors as obtained during the previous step were used, resulting in images of axial, radial, and diffusivity as well as fractional anisotropy (FA). Spatial normalization to MNI space was carried out with tract-based spatial statistics[91] by creating a white matter skeleton from the mean FA image and subsequent mapping of individual FA images. The obtained transformations were applied to the three diffusivity metrics.

**Statistics and reproducibility**. All statistical tests were two-sided and the reported $p$-values were corrected for multiple comparisons. For imaging data, this was realized with family-wise error correction (MCM and gray matter volume: $p < 0.05$ FWE cluster level, following $p < 0.001$ uncorrected voxel level in SPM12; white matter microstructure: $p < 0.05$ FWE-corrected with threshold-free cluster enhancement, 500 random permutations in FSL). For behavioral data, head motion, and all post-hoc tests, the corrections were done with the Bonferroni-Holm procedure (number of performed tests for each analysis given below in brackets).

Behavioral data acquired during the execution of Tetris® was defined as the score per minute, allowing a comparison between task performance during the PET/MRI scans and the online training. For the mental rotation task, the overall duration to solve all image pairs (low = good performance) divided by the number of correct answers (high = good performance) was used as a summary measure. For the Tower of London and match-to-sample tasks, the processing time and the number of correct answers were used as the outcome, respectively. Behavioral data were analyzed with a repeated-measures ANOVA in Matlab with factors group

(control, training), time (PET/MRI scan 1 and 2), and condition (easy, hard). Training-induced changes were assessed by the interaction effect group*time*condition (one interaction). In the post-hoc analysis the interaction group*time was tested for each condition separately (two conditions), followed by paired t-tests of the time effect within each group (two groups). Independent-samples t-tests were used to assess differences in initial task performance between the training and control group (two conditions). For the training group, additional paired t-tests were conducted to test for differences in Tetris® score between the second PET/MRI scan and the final visit (two conditions).

Imaging data were evaluated in a similar manner, testing for training-induced changes in MCM. First, a repeated-measures ANOVA was implemented in SPM12 with factors group (control, training), time (PET/MRI scan 1 and 2), and condition (rest, easy, hard). As SPM allows only two factors (in addition to the subject factor), the factor time was implicitly included by entering the difference maps between PET/MRI scan 2 vs 1 into the model. Based on our experimental design and the subsequent expectation that the hard task level will show the strongest neuroplastic changes, we tested for a conventional interaction effect. However, considering the previous similarity of MCM effects between the easy and the hard task conditions[19] the second contrast of interest was chosen as rest vs. (easy + hard). For the subsequent post-hoc analysis regional MCM values were extracted from the significant clusters and analyzed in Matlab. Here, the interaction group*time was tested for each condition separately (three conditions), followed by paired t-tests of the time effect within each group (two groups).

MCM values at measurement 2 (i.e., after training) were correlated with the corresponding behavioral data. Based on our findings (see results), the divergence between MCM increases at rest and decreases during the task can be summarized by their difference. Normalization by the mean value further takes the absolute value into account while avoiding mathematical coupling[92]. The MCM difference was also correlated with the overall learning performance, quantified by the normalized area under the curve of scoring obtained during the entire 4-week training period. Spearman's rho was calculated to account for one outlying value.

Structural changes in gray matter volume and white matter microstructure were calculated by repeated-measures ANOVA in SPM12 and FSL, respectively. The analysis included the factors group (control, training) and time (PET/MRI scan 1 and 2), testing for their interaction.

Differences in head motion (i.e., framewise displacement) during FC acquisition were assessed by repeated-measures ANOVA in Matlab with factors group (control, training), time (PET/MRI scan 1 and 2), and condition (rest, easy, hard). Again, we tested the interaction effect group*time*condition (one interaction) and post-hoc interactions group*time for each condition separately (three conditions).

To verify the reproducibility of the results at the individual level, the first PET/MRI measurements of the first 22 subjects were partly analyzed twice by two different investigators. At the group level all statistical tests were reproduced again after 4 months to confirm the initial findings by the same person originally conducting the analyses. All mentioned attempts at replication were successful.

**Simulations**. Simulations were carried out to identify whether the training-specific changes in MCM were driven by CMRGlu or FC. As MCM represents the spatial correlation between voxel-wise patterns of CMRGlu and FC, voxels in the target region were removed in a step-wise manner and the training effects (i.e., F-value of group*time interactions) were recalculated. Thus, a decrease in the training effect is expected if the observed changes in MCM are indeed driven by one of the parameters.

First, voxels of the target region were removed randomly and the resulting MCM z-scores were averaged for 500 random selections of voxels. Second, voxels were progressively removed based on the amplitude values of the two imaging parameters. More specifically, voxels were removed either on the basis of the underlying CMRGlu or FC values, starting with those voxels that contained the lowest values. For both the random and specific approaches, an increasing amount of voxels were removed in steps of 10% up to 90%.

**Reporting summary**. Further information on research design is available in the Nature Research Reporting Summary linked to this article.

## Data availability
Raw data will not be publicly available due to reasons of data protection. Processed data can be obtained from the corresponding author on request with a data-sharing agreement.

## Code availability
Custom code can be obtained from the corresponding author on request.

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

## Acknowledgements
We are particularly grateful to Prof. Danny JJ Wang, Stevens Neuroimaging and Informatics Institute, University of South California, and the Regents of the University of California, for providing the pCASL sequence. Stimulus images for the mental rotation task are courtesy of Michael J. Tarr, Carnegie Mellon University, http://www.tarrlab.org/. We thank the graduated team members and the diploma students of the Neuroimaging Lab (NIL, head: R. Lanzenberger) as well as the clinical colleagues from the Department of Psychiatry and Psychotherapy for clinical and/or administrative support. In detail, we would like to thank S. Kasper, K. Papageorgiou, P. Michenthaler, T. Vanicek, A. Basaran, M. Hienert, L. Silberbauer, J. Unterholzner, and G. Gryglewski for medical support, M. B. Reed for analysis support, V. Ritter, K. Einenkel, and E. Sittenberger for subject recruitment, and A. Jelicic for partly implementation of the task. We are further grateful to J. Raitanen, J. Völkle, and A. Pomberger for radioligand synthesis. This research was funded in whole, or in part, by the Austrian Science Fund (FWF) KLI 610, PI: A. Hahn. For the purpose of open access, the author has applied a CC BY public copyright license to any Author Accepted Manuscript version arising from this submission. S.K. is supported by the MDPhD Excellence Program of the Medical University of Vienna. L.R. and M.K. were recipients of a DOC Fellowship of the Austrian Academy of Sciences at the Department of Psychiatry and Psychotherapy, Medical University of Vienna. The scientific project was performed with the support of the Medical Imaging Cluster of the Medical University of Vienna.

## Author contributions
Study design: A.H., R.L., and M.H. Data acquisition: L.R., A.H., S.K., G.M.G., V.P., W.W., and M.K. Data analysis: A.H., L.R., and M.K. Manuscript preparation: S.K., G.M.G., and A.H. All authors discussed the implications of the findings and approved the final version of the manuscript.

## Competing interests
The authors declare the following competing interests: R.L. received travel grants and/or conference speaker honoraria within the last 3 years from Bruker BioSpin MR and Heel, and has served as a consultant for Ono Pharmaceutical. He received investigator-initiated research funding from Siemens Healthcare regarding clinical research using PET/MRI. He is a shareholder of the start-up company BM Health GmbH since 2019. M.H. received consulting fees and/or honoraria from Bayer Healthcare BMS, Eli Lilly, EZAG, GE Healthcare, Ipsen, ITM, Janssen, Roche, and Siemens Healthineers. W.W. declares to having received speaker honoraria from the GE Healthcare and research grants from Ipsen Pharma, Eckert-Ziegler AG, Scintomics, and ITG; and working as a part time employee of CBmed Ltd. (Center for Biomarker Research in Medicine, Graz, Austria). The remaining authors declare no competing interests.
