## [Peer Review File · Communications Biology]

Reviewers' comments:

Reviewer #1 (Remarks to the Author):

To author,

This manuscript evaluated hierarchical network interactions in the performance of a visuo-spatial processing task using simultaneous PET/MR imaging. This topic would be of interest to the neuroscientists exploring the neurobiological basis of learning. However, the narrative of the thesis is somewhat distracting, and there is a logical leap in reaching the conclusion. There are several concerns that need to be modified or supplemented in this manuscript. These are described below:

1) In this study, video game Tetris was used as a cognitive task for learning, and the brain region with mutual task-specific effects across imaging modalities was shown as the occipital cortex. These results may not appear in general for learning, but may appear only in learning specialized for visuospatial tasks. Please, discuss the limitations associated with this and incorporate them into your conclusions.

2) In this study, brain regions with mutual task-specific effects across imaging modalities comprised the occipital cortex, intraparietal sulcus, and frontal eye field, but the intraparietal sulcus, and frontal eye field was not used as the target region in evaluating learning-induced adaptations in metabolic connectivity mapping. It is need to provide additional analyzes or discuss reasons for not analyzing them.

3) In this study, to evaluate learning-induced adaptations in metabolic connectivity mapping, network changes after practicing the same task by computing the association between CMRGlu and BOLD-derived FC were investigated. The rationale for this should be discussed in depth in the discussion section.

4) The descriptions of the introduction and discussion are generally scattered and too much, making it difficult to understand the main research results and related issues. It seems that the main points need to be sorted out.

5) Throughout the paper, corrections are necessary for accurate representation.

ex 1) The group name is not used consistently (page 18; learning or passive control group, training or control group).

ex 2) There is no explanation for M1 and M2 in Figures 2 and 3.

Reviewer #2 (Remarks to the Author):

The study is a logical next step for this research team in the investigation of relationship between FDG PET and BOLD MRI during resting and activated state of brain functioning using their original approach. The basic methodology relies on performing quantitative (with arterial sampling for input function) FDG PET scan using continuous infusion of 18F-FDG in parallel with BOLD fMRI on PET/MR scan. Initial ~8 minutes of scanning are done at resting state (eyes opened fixed at crosshair) followed by 4 periods (6 min each, 2 easy and 2 hard) of task (playing adopted version of video game "Tetris") intermittently with periods of resting state (5 min length each) following tasks. Thus, the activation paradigm is more typical for fMRI studies, and it is applied to FDG scan assuming (based on the previous findings) that continuous inhalation technique allows careful separation of resting and activated states within one FDG PET imaging session.

The main strategy here was combining MRI-derived connectivity information with FDG-derived metabolic data to create "metabolic connectivity mapping". The focus of this project was to evaluate the effects of learning (4 months of active playing adopted Tetris game in active group and not doing it in the control group) on the resting state and activated (with the same Tetris game) metabolic connectivity. As the result of training, task performance improved (especially on hard task) and even stayed improved 4 weeks after the end of training. Moreover, training improved performance of several cognitive tests involving mental rotation and visual search, but

not spatial planning performance. Brain regions with increased CMRGlc, CBF (ASL) and BOLD during task served as regions for the assessment of learning-induced changes in metabolic connectivity. With occipital cortex as the target regions (the only one which was increasing in all modalities), learning-induced changes in metabolic connectivity were observed in the dorsal anterior cingulate cortex and insula (part of salience network). Post hoc analyses demonstrated that metabolic connectivity connections from dorsal anterior cingulate and insula toward occipital cortex increased at resting state, but decreased during execution of hard level of Tetris game in learning group compared to controls. These differences in metabolic connectivity values between the rest and hard task correlated with the Tetris score. Additional simulation analysis suggested that learning specific changes in metabolic connectivity in resting state were dependent on CMRGlc (and not BOLD), whereas in the hard task condition, they were driven by BOLD and not CMRGlc. Overall, the approaches and findings are of interest to the audience of the Communication Biology. The manuscript is well written, thoughtful and provide detailed information on the experiments and analyses, and exhaustive discussion. Findings are supported by proper illustrations and cited references from existing literature.

Few comments.

1. Four hours of fasting before FDG scan is rather short (typically, these are 5-6 hours). It will be helpful to provide blood glucose levels at the beginning of FDG scan, and confirm that they were not substantially different at the end of FDG infusion.

2. "...we combined the imaging parameters of glucose metabolism (CMRGlu), blood flow (CBF) and the BOLD signal for a functional delineation of brain regions with increased metabolic demands during task performance".

Why ASL (CBF) was used in addition to CMRGlc and BOLD? The whole story is about the relationship between CMRGlc and BOLD. Moreover, ASL was not involved in any of further analyses. Will the results be different if ASL is not used?

3. Intensive playing of video game (Tetris) improved performance of several mental tests including mental rotation and visual search, the effect stayed for 4 months after the end of training, and general resting and activated state estimates of brain metabolic activity and connectivity were modified. "Tetris" is a challenging game but not the only one, which some people play on the regular basis. Now younger adults spend a lot of time playing video games and some of these games are quite effortful. Did you check whether participants from control group do not play other mentally effortful games on the regular basis, which could change some psychometric, metabolic and functional MRI parameters? The control group did not improve on cognitive tests evaluated including mental rotation and visual search, however they could have other effects which may be important for brain functioning and metabolism. Moreover, one of the important next goals of this research is the investigation of effects of aging and neurodegenerative disorders. However, older adults (and especially symptomatic individuals) are usually not playing those "hard level" games. Please suggest on how this potential cohort effect could be controlled.

Reply to Reviewer's Comments

Legend:

Comment from reviewer.

Response to reviewer comment.

Text as it appeared in the previous manuscript version.

New text as it appears in the revised manuscript version.

Text removed from the manuscript but included in the response letter for convenience.

Page and line numbers refer to the revised manuscript.

Reviewer #1:

Remarks to the Author:

This manuscript evaluated hierarchical network interactions in the performance of a visuo-spatial processing task using simultaneous PET/MR imaging. This topic would be of interest to the neuroscientists exploring the neurobiological basis of learning. However, the narrative of the thesis is somewhat distracting, and there is a logical leap in reaching the conclusion. There are several concerns that need to be modified or supplemented in this manuscript. These are described below.

Response: We thank the reviewer for the valuable comments and the opportunity to further improve our manuscript. We have carefully revised the manuscript, in particular introduction and discussion sections, please see also our response to issues 1.3 and 1.4 below.

1.1. In this study, video game Tetris was used as a cognitive task for learning, and the brain region with mutual task-specific effects across imaging modalities was shown as the occipital cortex. These results may not appear in general for learning, but may appear only in learning specialized for visuospatial tasks. Please, discuss the limitations associated with this and incorporate them into your conclusions

Response: We agree with the reviewer that this requires further attention. We have now adapted the relevant sections of the text and included this in the limitation as detailed below. Moreover, we elaborate this aspect in section 1.4, during the revision of the introduction and discussion.

Abstract, page 3, lines 60-62

*Accordingly, a higher divergence between resting-state and task-specific effects was associated with better cognitive performance, indicating that these adaptations are complementary, and both required for successful **visuo-spatial** skill learning.*

Discussion, page 11, lines 249-252

*Together, these findings highlight **that** the interaction of both metabolically expensive general neuroplastic adaptations and task-specific network reorganizations is required for improvement in behavioral performance **of visuo-spatial skill learning.***

Conclusion, page 16, lines 384-391

*In sum, **applying MCM to simultaneous PET/MR imaging data of visuo-spatial learning enabled** the combined assessment of brain network dynamics and their underlying metabolic demands **at resting-state and during task performance.** Future studies may build on these findings by investigating if the described effects can be transferred to other learning paradigms or if they are specific for the visuo-spatial domain. Although we have carefully assessed the cognitive domains relevant for the Tetris® task, we acknowledge the possibility that further aspects may have an effect when investigating brain function longitudinally. However, testing these would have exceeded the scope of this work.*

1.2 In this study, brain regions with mutual task-specific effects across imaging modalities comprised the occipital cortex, intraparietal sulcus, and frontal eye field, but the intraparietal sulcus, and frontal eye field was not used as the target region in evaluating learning-induced adaptations in metabolic connectivity mapping. It is need to provide additional analyzes or discuss reasons for not analyzing them.

Response: We thank the reviewer for highlighting this aspect. We have indeed evaluated learning effects with all three regions as targets, i.e., FEF, IPS and Occ, but the former two regions did not show any significant training effects. We have made several changes to the text to make this clear.

Methods, page 25, lines 610-612:

The brain regions identified in this analysis were used as target regions for the subsequent MCM analysis [...].

Results, page 9, lines 187-188:

*Brain regions with mutual task-specific effects across imaging modalities (i.e., intersection) comprised the occipital cortex (Occ, 14.0 cm³), intraparietal sulcus (IPS, 20.3 cm³) and frontal eye field (FEF, 6.9 cm³), with the latter two representing the dorsal attention network (DAN). **These three areas served as target regions** for the subsequent assessment of learning-induced changes with MCM.*

Results, page 10, lines 210-211:

*Furthermore, no significant training-induced effects in MCM were observed for the other two **target** regions that showed overlapping task-specific activation (i.e., **FEF and IPS**).*

Discussion, page 11, lines 258-261

*However, the DAN **represented by FEF and IPS** showed no relevant training effects, indicating that this network mirrors a more general involvement in visuo-spatial processing required for planning and problem solving of the current task.*

1.3. In this study, to evaluate learning-induced adaptations in metabolic connectivity mapping, network changes after practicing the same task by computing the association between CMRGlucose and BOLD-derived FC were investigated. The rationale for this should be discussed in depth in the discussion section.

Response: We apologize for the lack of clarity regarding the specific opportunities arising from the framework of MCM. To convey these aspects more clearly, we have adapted the text in several positions. For convenience, we have also indicated the text that was removed. However, we would also like to emphasize that an in-depth discussion for computing the association between BOLD-derived functional connectivity and glucose metabolism has already been provided in previous work from others [Riedl et al., 2016] and our own group [Hahn et al. 2020]. To avoid a repetition, we highlight the most compelling aspects in the introduction of the current manuscript and kindly refer the reader to previous work for more details.

Introduction, pages 4-5, lines 85-125

*However, most of the previous work only employed a single imaging modality at the same time, thus impeding to draw conclusions about **how the** different parameters of brain function **act together in the process of learning**. In addition, **neuroplastic effects** were investigated either in a general manner at resting state (e.g., gray and white matter structure¹, network adaptations) or specifically during task execution (e.g., metabolic demands⁶, neuronal activation), while the direct comparison between **the two states** largely remains missing⁷. In sum, it is not clear whether intrinsic resting-state or task-related effects drive the improvement in cognitive performance after learning. Furthermore, the interaction between different indices of brain function and network adaptations is poorly understood.*

The application of functional PET (fPET)⁸ in the framework of metabolic connectivity mapping (MCM) provides a valuable approach to address both of these open questions. MCM combines MRI-derived functional connectivity and glucose metabolism obtained with [¹⁸F]FDG PET, thereby enabling the computation of directional connectivity⁹. The underlying rationale is that the integration of metabolic information identifies the target region of a connection, since the majority of energy demands emerge post-synaptically¹⁰⁻¹². The two imaging parameters are also tightly linked on a physiological basis through glutamate-mediated processes that occur upon neuronal activation. Glutamate release increases cerebral blood flow via neurovascular coupling^{13,14}, which in turn affects the blood oxygen level dependent (BOLD) signal used for the assessment of functional connectivity. On the other hand, glutamate release also triggers glucose uptake into neurons¹⁵ and astrocytes¹⁶, to meet increased energy demands for the reversal of ion gradients^{17,11,18}. MCM thus constitutes a validated framework to investigate the associations of glucose metabolism and functional connectivity and decipher hierarchical interactions across brain regions by assigning directionality to connections. For an in-depth discussion on the rationale and the underlying biological mechanisms of MCM the reader is referred to previous work^{9,19}. Furthermore, the use of functional PET (fPET) imaging allows to investigate metabolic demands at rest and during task execution in a single measurement¹⁹. Combining these methodological advancements we have recently Using MCM, we have recently demonstrated that first-time performance of a cognitive task strengthened the interplay of functional connectivity and glucose

metabolism, specifically for feedforward connections to higher-order cognitive processing areas¹⁹. ~~These data indicated that most of the metabolic cost originates from the switch of the resting state to the task-related network interactions, which extended previous work showing that acute task performance itself leads to pronounced functional network reorganizations²⁰ and increases in metabolic demands²¹. However, the corresponding effects induced by prolonged training of a task remain unknown.~~

In the current work we aimed to address the open questions outlined above, namely i) the interaction of training-induced changes between functional connectivity and glucose metabolism, ii) the neurobiological contributions of resting-state and task-specific effects that drive improvements in cognitive performance and iii) the hierarchical interplay across brain regions involved in the learning process. We investigated learning-induced neuronal adaptations in functional brain networks and the underlying energy demands with MCM before and after healthy volunteers practiced a challenging visuo-spatial task for 4 weeks. Proceeding from the convergence of functional connectivity and glucose metabolism already during the first execution of a novel task¹⁹ we expect that after continuous skill learning this task-specific association is consolidated also at resting-state.

1.4 The descriptions of the introduction and discussion are generally scattered and too much, making it difficult to understand the main research results and related issues. It seems that the main points need to be sorted out.

Response: We thank the reviewer for the valuable remark that the main points of the introduction and the discussion need to be emphasized to improve comprehensibility and clarity. We therefore accentuated the main points, shortened explanations and rephrased the transitions between sections. Furthermore, we aimed to improve readability by moving paragraphs and clearly separating our interpretation of the underlying neurobiological mechanisms from the other sections. Again, text that was removed is shown in the response letter for convenience. Please see our response to issue 1.3. for detailed changes of the introduction.

Discussion, page 11, lines 238-242:

We employed brain network analyses of simultaneous PET/MR imaging to investigate learning-induced neuroplastic changes in functional network reorganization and the underlying metabolic demands that relate to cognitive performance improvements. MCM served as a suitable multimodal approach to assess task-specific and resting-state adaptations, which earlier have been examined independently.

Discussion, pages 11-12, lines 263-267:

~~*As part of the SN, the right anterior insula mediates switching between task-irrelevant networks and the activation of task-specific networks that convey externally oriented attention²⁷⁻³⁰. For instance, visual stimuli can be processed by the anterior insula through the dorsal visual pathway and the intraparietal sulcus^{28,31}. The dorsal visual pathway thus represents a crucial bottom-up connection to the SN. Moreover, the anterior insula functions as an essential hub to filter relevant bottom-up stimuli and to facilitate access to working memory resources, explaining its involvement in multiple sensory and cognitive fields^{28,32,33}.*~~

*The right anterior insula mediates switching between task-irrelevant and task-specific networks that convey externally oriented attention²⁵⁻²⁸ through the dorsal visual pathway and the intraparietal sulcus^{26,29}. **This pathway thus represents a crucial bottom-up connection to the SN with the anterior insula as a feedforward stimuli filter and working memory access hub^{26,30,31}.***

Discussion, page 12, line 268-272: (moved up one paragraph)

~~*Generally, the dACC plays an essential role in monitoring cognitive control by adjusting automatic behavioral patterns to meet specific task demands^{39,40,35}. In more detail, dACC activation is associated with occurring conflicts between control signals^{39,41-43}, error monitoring⁴⁴⁻⁴⁶ and negative feedback⁴⁷. Inhibitory transcranial direct current stimulation of the dACC leads to decreased neuronal responses following errors and negative feedback, which subsequently results in poorer accuracy and reduced learning⁴⁸. These essential cognitive functions have led to an integrative account of dACC function to provide task representations⁴⁵. Optimization of task representations through adaptive coding of task-relevant variables enables cognitive control and guidance of behavior. It was further proposed that the adjustments of dACC connections represent a flexible process, emphasizing its role in learning⁴⁵.*~~

On the other hand, the dACC plays an essential role in cognitive control³²⁻³⁷ error monitoring³⁸⁻⁴⁰ and negative feedback⁴¹, whereas inhibition of dACC functioning results in poorer accuracy and reduced learning⁴². These cognitive aspects have led to an integrative account of dACC function to provide **optimized** task representations through adaptive coding of task-relevant variables, which further guide behavior³⁹.

Discussion, page 12, line 273-284: (moved down one paragraph)

Furthermore, the anterior insula and dACC were both identified to exert activation patterns preceding task errors⁴³ and were associated with performance monitoring³⁵. Since training led to a marked improvement in task performance, the associated decrease of insula and dACC input to Occ during task execution might indicate a reduced error rate. The observation of changes in cognitive performance and MCM mainly for the hard task condition is well in line with the experimental design of our Tetris® task, where only the hard level was set to require specific training. In accordance, a previous study showed that activation of the SN is dependent on cognitive load^{36,37}; other reports^{44,45} and the experimental design of our Tetris® task, where only the hard level was set to require specific training. The accompanying behavioral data further suggest that improvements in task performance were specific to abilities of mental rotation and, to a lesser extent, visual search and working memory, but not spatial planning and problem solving. Following the assumption that mental rotation reflects a continuous transformation performed on visual representations in the human brain⁴⁶, we speculate that the observed improvement is based on the representational adaptation described below.

Discussion, page 13, lines 296-299:

~~*We observed adaptations of MCM at both resting-state and task execution. Although these effects diverged they seem to constitute complementary adaptations of the learning process. We thereby provide a unified interpretation of skill learning as an advancement of neuronal representations of the task.*~~

The hierarchical interaction across brain regions as assessed with MCM revealed divergent yet complementary effects of the learning process at resting-state and during task execution. This approach further enabled us to provide a unified interpretation of skill learning as an advancement of regionally specific neuronal representations of the task.

Discussion, page 13, lines 306-308:

The initial learning stage comprises interactions with unknown task demands and high perceptual load, thus requiring adaptation of attention due to limited processing resources^{49,50}.

Discussion, pages 13-14, lines 311-321:

~~*The attentional optimization is implemented by recurrent loops between across different hierarchical levels of cortical systems^{56,57}, based on synaptic gain or responsiveness of post-synaptic neurons that encode prediction errors^{51,58}. Within these loops, Here, so-called state and error units characterize the different representations and their inaccuracy (i.e., the prediction error), respectively⁴⁸. The prediction error itself is optimized through top-down modulatory control⁴⁸, e.g., by the SN²⁶. Thus, attention selects⁶¹⁻⁶³ and sharpens⁵² the representations that enter working memory. Subsequently, cognitive control enables to distinguish between efficient and unsuccessful task rules, thereby implementing an advanced representation of the task^{52,64}, which enables to select^{49,59,60} and sharpen⁵⁰ representations and to*~~

distinguish between efficient and unsuccessful task rules, thereby implementing an advanced representation of the task^{50,61}. Training will continuously alter the task representation by incorporating relevant and discarding irrelevant information, approaching an optimal solution and finally yielding a sparse representation^{62,63}. encoded only by a small portion of neurons⁶⁵. This may enable faster processing through inhibition of neurons that encode irrelevant features⁶⁶.

Discussion, page 14, lines 339-342:

~~By providing a neurobiological explanation of our findings, we acknowledge that these interpretations rely on previous work mostly obtained from preclinical studies.~~
In this study, the complementary task- and resting-state learning effects were identified by simultaneous PET/MR imaging. While interpreting our results at the neurobiological level, we acknowledge that the following section relies on previous work mostly obtained from preclinical studies.

Discussion, page 15, lines 349-353:

~~Repeated tagging through continuous training of the task then induces synaptic capture, where plasticity-related products (e.g., ARC, AMPA receptor subunit GluR1) are mobilized to stabilize the spine architecture. Furthermore, the sharpening of representations might involve the stabilization process of synaptic clustering, i.e., the addition of new spines to dendritic sites that have been captured during earlier training session^{74,65,75,76}. These structural adjustments enable functional synaptic potentiation by anchoring additional postsynaptic AMPA receptors⁷¹, finally representing the consolidated skill engram.~~

Discussion, page 15, lines 363-371:

~~Recent metabolic simulations of synaptic adaptations supported the neurobiological process of synaptic tagging and capture as a plausible physiological mechanism to increase energy efficiency^{66,74}. It is based on a molecular optimization where a vast space of potential synaptic connections is explored to find the most appropriate representation in an energy efficient manner⁶⁹. Computations emphasized that storage of transient memories without protein synthesis is up to ten-fold more energy efficient, compared to the immediate formation of long-term potentiation that requires a high amount of metabolic resources⁷⁴. This is in line with our own simulations, indicating that energy demanding (synaptic capture) learning effects at rest were dependent on CMRGlu, while transient storage of information (synaptic tagging) elicited by task execution was driven by FC. Combining the theories above, we speculate that the simulated removal of voxels based on CMRGlu equals a perturbation of state units that represent the skill engram at resting-state. We therefore speculate that simulations affecting CMRGlu equal a perturbation of state units that represent the skill engram at resting-state.~~

Discussion, page 16, lines 377-381: (moved up from section “Limitations, outlook and conclusions”)

We are aware that causal relationships between the observed findings from human brain imaging and more invasive work underpinning the prediction error model as well as the process of synaptic tagging and capture still need to be established. Although these models can be unified in our data, future work should aim to resolve this knowledge gap by directly testing the suggested associations.

Conclusion, page 16, lines 384-391:

In sum, applying MCM to simultaneous PET/MR imaging data of visuo-spatial learning enabled the combined assessment of brain network dynamics and their underlying metabolic demands at resting-state and during task performance. Future studies may build on these findings by investigating if the described effects can be transferred to other learning paradigms or if they are specific for the visuo-spatial domain. Although we have carefully assessed the cognitive domains relevant for the Tetris® task, we acknowledge the possibility that further aspects may have an effect when investigating brain function longitudinally. However, testing these would have exceeded the scope of this work.

1.5. Throughout the paper, corrections are necessary for accurate representation.

ex 1) The group name is not used consistently (page 18; learning or passive control group, training or control group).

ex 2) There is no explanation for M1 and M2 in Figures 2 and 3.

Response: Thank you for the attentive remarks. We have adapted the text accordingly, so that “learning group” now consistently reads “training group”, and “passive control group”, as “control group” only. Moreover, we have pointed out that M1 and M2 refer to first and second measurement and scanned the manuscript once again for typos.

Results, page 7, lines 129-130:

Subjects were assigned to either the training (n=21) or ~~passive~~ control group (n=20) [...].

Methods, page 18, line 411:

*In this longitudinal study participants were randomly assigned to the **training** or ~~passive~~ control group [...].*

Figure 1, page 36, line 998:

After the initial screening, participants were randomly assigned to the training or the ~~passive-control~~ group.

Figure 2, page 38, lines 1025-1026:

*Changes in task performance differed between the two PET/MRI measurements (**M1 and M2**), groups and task conditions (group*time*condition interaction, $p < 10^{-5}$).*

Figure 3, page 39, lines 1052-1053:

*There were no significant changes in the control group between the two measurements (**M1, M2**).*

Reviewer #2:

Remarks to the Author:

The study is a logical next step for this research team in the investigation of relationship between FDG PET and BOLD MRI during resting and activated state of brain functioning using their original approach. The basic methodology relies on performing quantitative (with arterial sampling for input function) FDG PET scan using continuous infusion of 18F-FDG in parallel with BOLD fMRI on PET/MR scan. Initial ~8 minutes of scanning are done at resting state (eyes opened fixed at crosshair) followed by 4 periods (6 min each, 2 easy and 2 hard) of task (playing adopted version of video game “Tetris”) intermittently with periods of resting state (5 min length each) following tasks. Thus, the activation paradigm is more typical for fMRI studies, and it is applied to FDG scan assuming (based on the previous findings) that continuous inhalation technique allows careful separation of resting and activated states within one FDG PET imaging session.

The main strategy here was combining MRI-derived connectivity information with FDG-derived metabolic data to create “metabolic connectivity mapping”. The focus of this project was to evaluate the effects of learning (4 months of active playing adopted Tetris game in active group and not doing it in the control group) on the resting state and activated (with the same Tetris game) metabolic connectivity. As the result of training, task performance improved (especially on hard task) and even stayed improved 4 weeks after the end of training. Moreover, training improved performance of several cognitive tests involving mental rotation and visual search, but not spatial planning performance. Brain regions with increased CMRGlc, CBF (ASL) and BOLD during task served as regions for the assessment of learning-induced changes in metabolic connectivity. With occipital cortex as the target regions (the only one which was increasing in all modalities), learning-induced changes in metabolic connectivity were observed in the dorsal anterior cingulate cortex and insula (part of salience network). Post hoc analyses demonstrated that metabolic connectivity connections from dorsal anterior cingulate and insula toward occipital cortex increased at resting state, but decreased during execution of hard level of Tetris game in learning group compared to controls. These differences in metabolic connectivity values between the rest and hard task correlated with the Tetris score. Additional simulation analysis suggested that learning specific changes in metabolic connectivity in resting state were dependent on CMRGlc (and not BOLD), whereas in the hard task condition, they were driven by BOLD and not CMRGlc.

Overall, the approaches and findings are of interest to the audience of the Communication Biology. The manuscript is well written, thoughtful and provide detailed information on the experiments and analyses, and exhaustive discussion. Findings are supported by proper illustrations and cited references from existing literature.

Response: We thank the reviewer for the thorough evaluation of our manuscript and the encouraging feedback on our work.

2.1. Four hours of fasting before FDG scan is rather short (typically, these are 5-6 hours). It will be helpful to provide blood glucose levels at the beginning of FDG scan, and confirm that they were not substantially different at the end of FDG infusion.

Response: We agree with the reviewer that further details on blood glucose levels are required. The minimum fasting time of 4 hours was actually set to the arrival at the university hospital. Notably, until the application of the radioligand another 1.5 h passed by, which yields a minimum fasting time of 5.5 h. We have provided blood glucose levels before the PET/MRI scan in the manuscript, however, we did not obtain these values after the PET/MRI. Usually, this is not acquired in PET experiments, since the radioligand should not change blood glucose levels due to different metabolism and the low amount of injected radioligand (μg range).

Methods, page 22, lines 520-521:

*Participants had to fast for at least **5.5 hours** before the start of the PET/MRI scan (except for unsweetened water), according to [^{18}F]FDG procedure guidelines⁸⁴.*

Methods, page 22, lines 539-540:

*Blood glucose levels were assessed right before the PET/MRI scan ($\text{Glu}_{\text{plasma}}$, triplicate) and were **5.45 ± 0.73 mmol per l**.*

2.2 “...we combined the imaging parameters of glucose metabolism (CMRGlu), blood flow (CBF) and the BOLD signal for a functional delineation of brain regions with increased metabolic demands during task performance”.

Why ASL (CBF) was used in addition to CMRGlc and BOLD? The whole story is about the relationship between CMRGlc and BOLD. Moreover, ASL was not involved in any of further analyses. Will the results be different if ASL is not used?

Response: We thank the reviewer for the interesting remark. The rationale to use all three imaging modalities was to ensure comparison to our previous work and to maximize the specificity regarding the functional definition of the target region. The BOLD signal represents a complex composite from several sources. On the other hand, ASL and fPET measurements provide more straightforward estimates of CBF and CMRGlu, respectively. The combination of BOLD, CBF and CMRGlu thus seems to provide the most robust functional delineation of target regions. We acknowledge that this approach may be conservative. Still, the trimodal combination ensures to include only voxels in the target region which are truly activated from as many neurobiological measurements as possibly obtained in the current study.

To provide a full picture of our results we have repeated the calculations when defining the target region without CBF, as suggested by the reviewer. As expected from supplementary figure S2, the target region of the occipital cortex increased in size (by 29%), whereas the FEF and IPS were rather stable (increase by 4% and 0.2%, respectively). Importantly, this increased occipital target region did not change the main findings, i.e., interaction effects and post-hoc differences remained stable for the insula and dACC. This is now included in the manuscript and supplementary figure S3.

Methods, page 25, lines 602-605:

*The three different indices of task-specific metabolic demands (CMRGlu, CBF, BOLD) were combined to obtain a robust estimate of regions involved in task processing. **This approach was chosen to enable comparison to our previous work¹⁹ and to maximize the specificity of the MCM target regions. Still, we also compute the main MCM training effects for the combination of CMRGlu and BOLD only (see supplementary figure S3).***

Results, pages 9-10, lines 208-210:

Moreover, the results remained stable when defining the target region only from task-specific CMRGlu and BOLD changes (i.e., without CBF, supplementary figure S3).

Supplementary figure S3: Learning-induced changes in MCM when defining the occipital cortex as target region from the intersection between task-specific CMR_{Glu} and BOLD signal changes. Compared to figure 3 (i.e., the target region defined from CMR_{Glu}, BOLD and CBF), these results were slightly weaker but still remained significant. Boxplots show the MCM z-scores of the insula and dorsal anterior cingulate cortex (dACC). Post-hoc comparisons indicate significant differences for the group*time interaction (^{##} $p=0.06$, [#] $p<0.05$, ^{##} $p<0.01$) and for the differences between the two measurements (^{*} $p<0.05$, ^{} $p<0.01$), corrected for multiple comparisons with the Bonferroni-Holm procedure. M1 indicates the first PET/MRI measurement and M2 the second, with training group in red ($n=21$) and control group in blue ($n=20$). Boxplots indicate median values (center line), upper and lower quartiles (box limits) and 1.5*interquartile range (whiskers).**

2.3. Intensive playing of video game (Tetris) improved performance of several mental tests including mental rotation and visual search, the effect stayed for 4 months after the end of training, and general resting and activated state estimates of brain metabolic activity and connectivity were modified. “Tetris” is a challenging game but not the only one, which some people play on the regular basis. Now younger adults spend a lot of time playing video games and some of these games are quite effortful. Did you check whether participants from control group do not play other mentally effortful games on the regular basis, which could change some psychometric, metabolic and functional MRI parameters?

Response: We agree with the reviewer that Tetris® is one of many video games that young participants might play regularly. Being well aware of this critical point, we instructed participants not to play and especially not to learn any (new) video games while participating in the study. Furthermore, with the exclusion criteria, we also controlled for similar video games mainly involving visuospatial skills like “Candy Crush” and others within the last three years.

Methods, page 21, lines 504-512:

Exclusion criteria were current and previous somatic, neurological or psychiatric disorders (12 months), substance abuse or psychopharmacological medication (6 months), current pregnancy or breast feeding, previous study-related radiation exposure (10 years), body weight of more than 100 kg for reasons of radiation protection, MRI contraindications and previous experience with the video game Tetris® within the last 3 years. Experience with and regular playing of similar video games, specifically games primarily involving visuospatial skills like “Candy Crush,” was another explicit exclusion criterium. Furthermore, participants of both groups were instructed not to play and especially not to learn any (new) video games while participating in the study.

2.4. The control group did not improve on cognitive tests evaluated including mental rotation and visual search, however they could have other effects which may be important for brain functioning and metabolism.

Response: We acknowledge the possibility that other effects might have evolved in the control group between the two PET/MRI scans. As Tetris® mainly involves visuospatial skills, we aimed to specifically test for the involved skill domains like mental rotation and visual search. Testing for other effects would have exceeded the scope of our study. We, therefore, included a corresponding remark in our limitations that the investigated effects were confined to the most relevant cognitive domains.

Limitations, outlook and conclusions, page 16, lines 388-391:

Although we have carefully assessed the cognitive domains relevant for the Tetris® task, we acknowledge the possibility that further aspects may have an effect when investigating brain function longitudinally. However, testing these would have exceeded the scope of this work.

2.5. Moreover, one of the important next goals of this research is the investigation of effects of aging and neurodegenerative disorders. However, older adults (and especially symptomatic individuals) are usually not playing those “hard level” games. Please suggest on how this potential cohort effect could be controlled.

Response: We can follow the reviewers concerns that older individuals or patients suffering e.g., from neurodegenerative disorders may experience difficulties to perform cognitively challenging tasks such as Tetris®. However, easier tasks that involve similar cognitive skills are already available in the clinical routine for cognitive training of the specific patient cohorts. Often, even only one button is needed to complete these tasks. Regarding the use of more complex tasks like Tetris®, we would also consider it feasible to assess individual cognitive abilities before the scan and adapt the task according to the respective individual abilities. This would also allow comparison between the aged and younger individuals.

Limitations, outlook and conclusions, page 16, lines 397-401:

*Disentangling the metabolic and functional requirements for neuroplasticity might prove beneficial to differentiate between different forms of neurodegenerative diseases and evaluate the severity of tissue damage in traumatic brain injury, **feasibly with the use of cohort-specific or individually adapted tasks.***

References

8. Hahn, A. *et al.* Quantification of Task-Specific Glucose Metabolism with Constant Infusion of ¹⁸F-FDG. *J. Nucl. Med. Off. Publ. Soc. Nucl. Med.* **57**, 1933–1940 (2016).
9. Riedl *et al.* Metabolic connectivity mapping reveals effective connectivity in the resting human brain PNAS 113: 428-433, 2016.
19. Hahn *et al.* Reconfiguration of functional brain networks and metabolic cost converge during task performance. *eLife* 9: e52443, 2020.
84. Guedj, E. *et al.* EANM procedure guidelines for brain PET imaging using [¹⁸F]FDG, version 3. *Eur. J. Nucl. Med. Mol. Imaging* **49**, 632–651 (2022).

REVIEWERS' COMMENTS:

Reviewer #1 (Remarks to the Author):

To author,

This manuscript evaluated hierarchical network interactions in the performance of a visuo-spatial processing task using simultaneous PET/MR imaging. All parts of the manuscript were revised properly according to the comments. This study can provide meaningful information about hierarchical networks to researchers interested in learning-induced neuroplastic mechanisms.

Reviewer #2 (Remarks to the Author):

Overall, the manuscript improved from the previous version, no more suggestions.